# Squat Detection and Estimation for Railway Switches and Crossings Utilising Unsupervised Machine Learning

**Yang Zuo** *, **Jan Lundberg, Praneeth Chandran** and **Matti Rantatalo**

Operation and Maintenance Group, Luleå University of Technology, 97187 Luleå, Sweden
* Correspondence: yang.zuo@ltu.se

**Abstract:** Switches and crossings (S&Cs) are also known as turnouts or railway points. They are important assets in railway infrastructures and a defect in such a critical asset might lead to a long delay for the railway network and decrease the quality of service. A squat is a common rail head defect for S&Cs and needs to be detected and monitored as early as possible to avoid costly emergent maintenance activities and enhance both the reliability and availability of the railway system. Squats on the switchblade could even potentially cause the blade to break and cause a derailment. This study presented a method to collect and process vibration data at the point machine with accelerometers on three axes to extract useful features. The two most important features, the number of peaks and the total power, were found. Three different unsupervised machine learning algorithms were applied to cluster the data. The results showed that the presented method could provide promising features. The k-means and the agglomerative hierarchical clustering methods are suitable for this data set. The density-based spatial clustering of applications with noise (DBSCAN) encounters some challenges.

**Keywords:** railway; S&C; switch and crossing; sensor; accelerometers; vibration; squat defect; clustering; unsupervised; k-mean; DBSCAN; agglomerative hierarchical

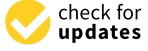



## 1. Introduction

Railway infrastructure is one key aspect of enabling economic success in a country [1]. Moreover, railway services must be highly reliable to attract more passengers and freight from other ways of transportation. Railway switches and crossings (S&Cs), are essential components of railway infrastructure. To be able to make the railway infrastructure reliable, critical components such as the S&C should be reliable because a single failure of it could lead to a large delay and huge economic loss. At the same time, since S&Cs are composed of diverse movable parts and can be regarded as a discontinuous point from the rail geometry point of view, they tend to experience higher failure rates compared with a normal piece of rail [2].

Maintenance for S&Cs is costly with the current application. A previous study performed by Cornish et al. [3] stated that S&Cs have occupied more than 24% of the total budget for all the maintenance activities against only 5% of the corresponding track length. In 2018 alone in Sweden, the S&Cs consumed 530 million SEK, which corresponds to around 10% of their whole maintenance budget [4]. S&C failures also cause significant delays in the railway system. Network Rail reports that S&C failures caused 481,719 delay minutes for the British railway network between 2019 and 2020 [5]. Only 5.5% of the total track length is S&Cs in Swedish railway system and caused 14% of all the delays [6]. Without taking proper maintenance actions, such failures could even lead to a fatal accident that would cause not only economic loss but also endanger the passengers and goods.

Due to the importance of S&Cs and their high maintenance costs, actions should be taken to prevent failures to happen. One approach to avoid failures is applying a condition-monitoring system to check the status of the S&C, using data analysis to provide

early warning before the failure and performing preventive maintenance. There are many methods that can be utilised to inspect the status of S&Cs.

One of the most common methods of rail and S&C inspection is applying manual on-site inspections, which means dedicated railway maintenance engineers needed to visit and inspect the rail at fixed intervals. They base their inspection on measuring tools, their vision, experience and insights to detect defects. However, this method is subjective and costly [7]. It sometimes can even put the inspectors in a dangerous situation [8]. Another way of inspecting the rails, including S&Cs, is using a dedicated track recording vehicle (TRV), where different optical sensors, accelerometers and gyro sensors are installed to measure different irregularities of the rails, according to European Standard EN 13848 [9]. This method might not be suitable for dedicated measuring an S&C [10]. It is both expensive and could even cause disruption to scheduled services [11]. Other non-destructive testing (NDT) techniques adopted to evaluate the rail defects include the vision-based techniques, ultrasound measurements, eddy current testing (ECT) systems, accelerometers, etc. [12]

The vision-based technique is one common method in current rail-defect detection studies [13,14]. The existing methods can be divided into two groups, namely, traditional image processing and image processing using machine learning. Yaman et al. [15] utilised the Otsu algorithm and extracted information about the rail surface. Then fuzzy logic was used to determine the defect types. Gan et al. [16] proposed an extractor to detect rail defects. Recently, with the popularity of deep convolution neural networks (DCNN), many researchers have started to apply deep learning to improve the vision-based techniques. Liang et al. [17] compared the SegNet algorithm with the manual and automatic threshold-segmentation algorithms and the results showed that the performance of the deep learning algorithm was 22.2% higher than manual threshold segmentation and was almost doubled compared with automatic threshold segmentation. Visual inspection is limited to detecting only the defects on the rail surface when they are not covered by snow or other obstacles. Dust, dirt and water on the lens can also decrease the detection accuracy.

The ultrasonic sensor measures and analyses the waves reflected back [18]. This method is suitable even for detecting the faults inside the rail (railhead and waist) [19]. Zhang et al. [20] proposed an ultrasonic testing method with a high-speed phased array. This multi-angle beam approach increases both the speed and the range of detection. Kaewunruen and Ishida [21] proposed a monitoring method using ultrasonic techniques to measure the crack propagation when forming squats. The results showed that the propagation can be approximated to be linear to the accumulated passing by tonnages up to some degree. The swift advancement of deep learning and other artificial intelligence (AI) technologies has led to significant progress in ultrasonic data processing [22]. This method needs accurate probe configurations and constant precise contact between the probe and the rail [23]. This makes the method suffer from technical challenges due to both environmental and operating conditions. Moreover, this technology faces challenges when detecting near-surface defects [23].

The ECT method is widely applied to inspect conductive material to analyse structural integrity. Alvarenga et al. [24] proposed a new system based on ECT for detecting and locating rail defects and they claimed an accuracy of 98%. The proposed method aimed to analyse eddy current data using wavelet transforms together with a convolutional neural network (CNN). Chandran et al. [25] proposed a differential eddy current sensor system on the train for fastener inspection. The proposed model achieved a detection accuracy of 96.79%. Kwon et al. [26] developed an advanced eddy current device with 16 channels to analyse the correlation between the depth and phase of a known artificial defect and later applied it to a natural defect specimen. The main challenge of the ECT system is the lift-off effect that affects the ECT signal, causing erroneous data interpretation [27].

Using vibration measurement is another popular method to detect rail defects. Vibrations usually can be measured from the train. Molodova et al. [28,29] proposed a method to improve the axle box acceleration (ABA) measurements for detecting light squats and performing health condition monitoring of insulated joints. One disadvantage of using

ABA is that the maximal ABA excited can be greater than 50 g, so a trade-off in sensitivity is needed. Wei et al. [30] proposed using the bogie acceleration (BA) measured for defect detection instead of ABA. A few advantages were emphasised. These two methods are both suitable for inspecting a long distance of rail.

One important type of rail defect is squats. Grosonni et al. [31] stated that almost a third of the recorded failures of the crossing panel were related to squats. Some studies have been performed about squat detection. Squat defects are the type of rail defect focused on in this study. Lesiak et al. [32] presented a simplified method for squat defect detection using laser scatterometry and the results achieved verified that it is effective in squat detection with both artificial and real defects. Ye et al. [33] utilised 3D reconstruction techniques to improve the performance and the achieved results demonstrated the possibility of using this technique for rail and crossing-nose inspection in railway systems. However, this type of method faces the challenges of the presence of water, dust and dirt on the laser lenses. Faghih-Roohi et al. [34] presented a DCNN approach to analyse images for rail surface defect detection and the experiments showed promising results. The results of DCNN architectures with different sizes and activation functions were tested to analyse the proposed method. However, the camera lens had similar problems as the laser transmitter. Kaewunruen et al. [21] proposed a monitoring system using ultrasonic techniques to measure the crack propagation when squats develop. The results showed that the propagation rate is roughly linear to the accumulated passing tonnages. This finding could help railway authorities, in advance, to plan more effective maintenance. However, the method of measuring the squat size were not described. Bocciolone et al. [35] investigated the feasibility of ABA-based rail status diagnosis. The signal-processing procedure produced some root mean square (RMS) band values and spotted one band level directly related to corrugation with short pitches. However, the algorithm was not implemented online to make real-time decisions, and this study did not establish the relationship between the level of the vibration amplitude and the depth of the corrugation with short pitches. Molodova et al. presented a few studies utilising ABA signals. These studies explored the influence of different parameters on the ABA signals [36] and tested an automated squat-detection system [37]. However, this approach aimed to perform squat detection for general rails, and it was not dedicated to performing condition monitoring for S&Cs. Moreover, the ABA signal could be influenced by various factors such as the type of axle box, the condition of the wheel axle bearings, and even the wheel profile. In addition, the vibration from both the bearing and the bearing defects would be mixed in the ABA measurements [38]. A fixed measurement system will be able to measure the status of an S&C more frequently, making it possible to detect defects at their early stage. Zuo et al. [39] proposed a method to utilise an unsupervised anomaly-detection technique to monitor the degree of squat defects. However, both time-domain and scale-averaged wavelet power (SAWP) features were needed.

This study investigated the possibility of utilising the data collected from three accelerometers mounted on a rod of the point machine and measuring the vibrations in three axes. Only features from the SAWP were extracted. The possibility of using data collected from three directions as separate test runs was investigated to increase the number of data points. Three unsupervised machine learning algorithms were utilised to identify different degrees of squat defects [10]. The studies performed by Li et al. [40,41] were from a different domain but the utilised techniques influenced the methodology part of this study.

## 2. Materials and Methods

### 2.1. The Testbed and Experiment Setup

This study presents a new alternative to extract features from vibration data collected at the S&C to utilise a few unsupervised machine learning techniques to cluster the data from two different S&Cs automatically. The testbed used a complete S&C and a bogie of 6 tonnes. The setup is similar to previous studies by Zuo et al. [39] and Zuo et al. [42]

The whole test site is visualised in Figure 1. The bogie used has two axles with a distance of 2.5 m in between. The angle of the S&C used is 1:16 and the total length is 38.14 m. The accelerometers were installed on a rod of the point machine. This approach provides an environment that secures and protects the embedded accelerometer. Another advantage of this setup is it provides easy access to the existing power supply. The vibration data in three directions and the speed of the bogie were measured and logged down. The testbed is illustrated and shown in Figure 2. The sensors 608A0 to 608A2 measure vibrations in the x,y and z directions, respectively . $S_0$ and $S_1$ are two ends of the rails with stop blocks in the through route. Manually introduced squats were named using the letters from A to K and the position of the point machine was located 5.86 m from $S_0$. To replicate two different levels of squat severity, the squats were manually created using a conventional angle grinder with depths of 1 mm and 4 mm. These depths represent distinct degrees of squat severity and were introduced intentionally to simulate surface squat defects. The measurements of the squats with these two different levels are shown in Table 1.

**Table 1.** Measurements of the squats.

| Squat | Squat Diameter Stage 1 (mm) | Max Depth Stage 1 (mm) | Squat Diameter Stage 2 (mm) | Max Depth Stage 2 (mm) |
|:---:|:---:|:---:|:---:|:---:|
| A | 43 | 1.2 | 62 | 3.7 |
| B | 41 | 1.0 | 61 | 3.9 |
| C | 42 | 1.0 | 63 | 3.7 |
| D | 42 | 1.0 | 66 | 4.4 |
| E | 0 | 0 | 65 | 3.7 |
| F | 42 | 1.1 | 65 | 4.2 |
| G | 42 | 1.0 | 64 | 3.7 |
| H | 42 | 1.5 | 62 | 4.7 |
| I | 42 | 1.4 | 62 | 4.3 |
| J | 42 | 1.2 | 63 | 4.4 |
| K | 42 | 1.1 | 61 | 4.1 |

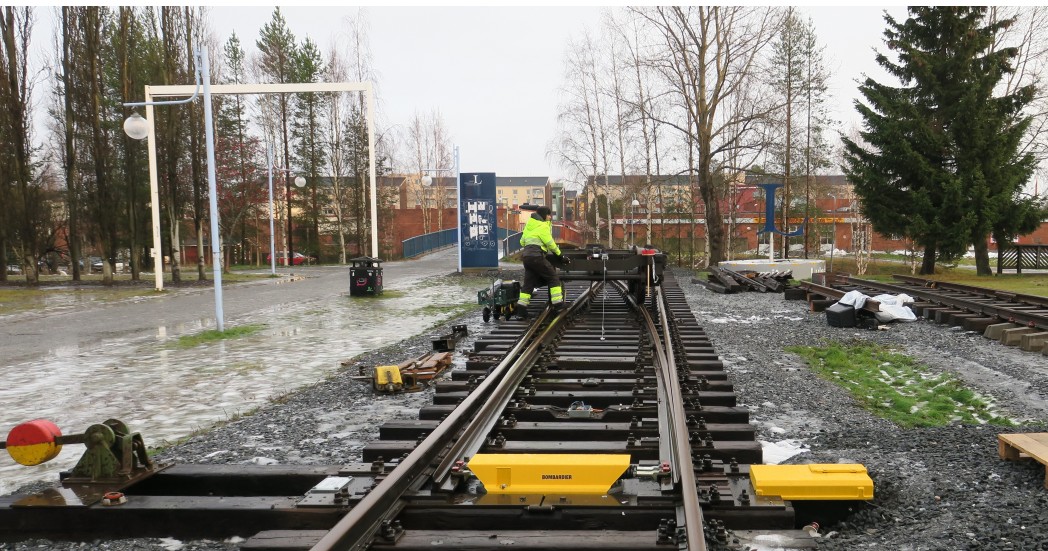

**Figure 1.** Experiment site.

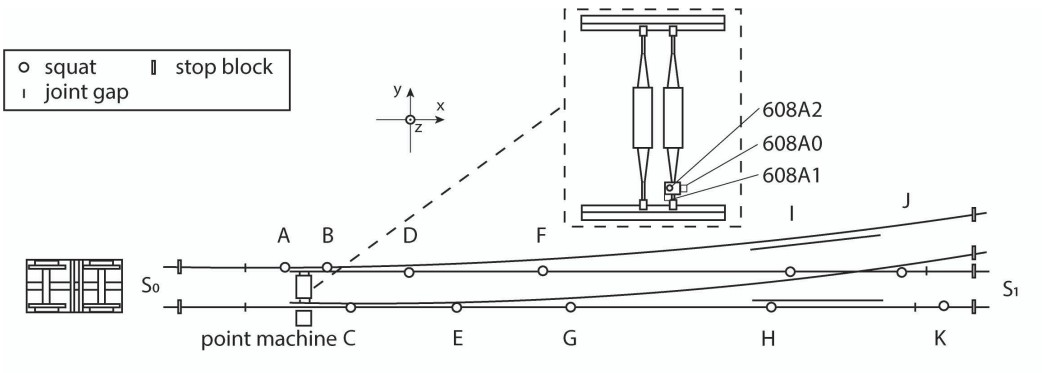

**Figure 2.** Diagram of the testbed and sensors placement.

### 2.2. Sensors

The accelerometers used in this experiment are IMI sensors model 608A. Some specifications of the sensors are listed in Table 2. This type of low-cost, general-purpose industrial accelerometer is suitable for trial test purposes and could be scaled up in the future. The installation locations of the sensors are illustrated in Figure 2. The actual installation in the testbed is visualised in Figure 3.

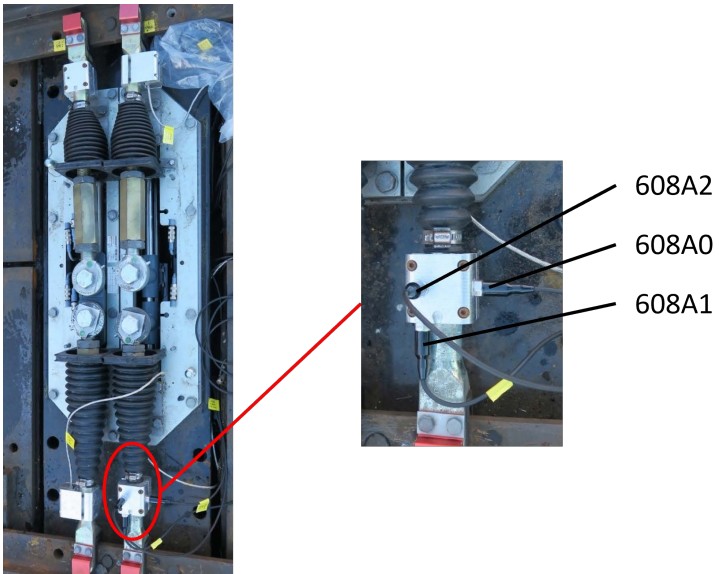

**Figure 3.** Sensors mounted on the point machine in the testbed.

**Table 2.** Parameters of the accelerometer used.

| Name | Range (Hz) | Sensitivity (mV/g) | Destruction Limit (g) | Resonant Frequency (kHz) |
|---|---|---|---|---|
| 608A | 0.5–10,000 | 10.2 | 50 | 22 |

### 2.3. Test Procedure and Data Acquisition

Three test scenarios were performed as follows. In each scenario, the bogie was travelling from the end labelled $S_0$ to the one labelled $S_1$. In the first scenario, the rails were without any squats. In the second scenario, 1 mm depth squats were generated. In the third scenario, the depth of the squat was increased to 4 mm. Every test scenario was run three times. The vibration signal was measured with three accelerometers installed on one rod of the point machine. To capture the vibration signals from all three accelerometers, the DAQ9174 data-acquisition platform was used to feed the data directly to a computer for storage. The measuring system operated at a sampling frequency of 51.2 kHz. The speed of

the bogie was measured using a customised tachometer consisting of a Hall effect sensor A3144 and multiple neodymium magnets. An Arduino Uno unit was used to transmit the revolutions per minute (RPM) of one of the front wheels to the computer via WiFi. The signals collected from a 4 mm test case are visualised in Figure 4.

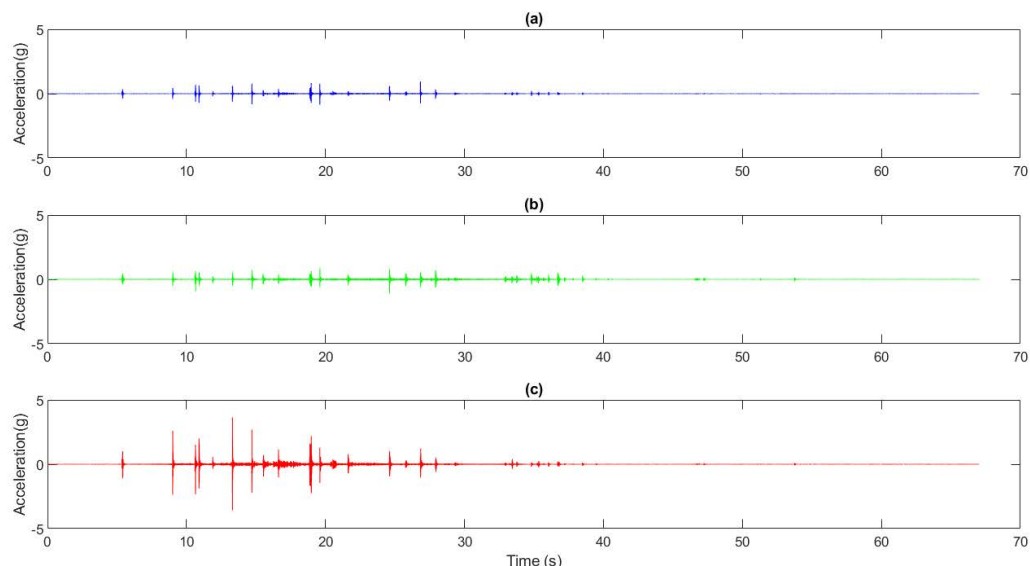

**Figure 4.** Amplitude comparison for signals from three different directions: (**a**) x direction. (**b**) y direction. (**c**) z direction.

### 2.4. Data Processing Procedure

Figure 5 describes the procedure of processing the measured signals. Before analysis, the vibration signals from each axis were standardized by comparing their maximum amplitudes. To further refine the data and remove any low-frequency noise or other unwanted signal components, a third-order Butterworth high-pass filter with a cutoff frequency of 100 Hz was applied to the signals. After standardization and filtering, the vibration signals were subjected to wavelet denoising using a Symlet 4 mother wavelet and level 9 decomposition. The denoising method employed the empirical Bayesian technique with a level-dependent noise estimator. Subsequently, a SAWP time series was derived from the processed signal. Finally, a set of features was extracted from the SAWP. The two most important features were selected after analyzing the Laplacian score. These features were then utilized in unsupervised machine learning clustering algorithms to group similar signals together. This approach can assist in identifying patterns or anomalies in the vibration data.

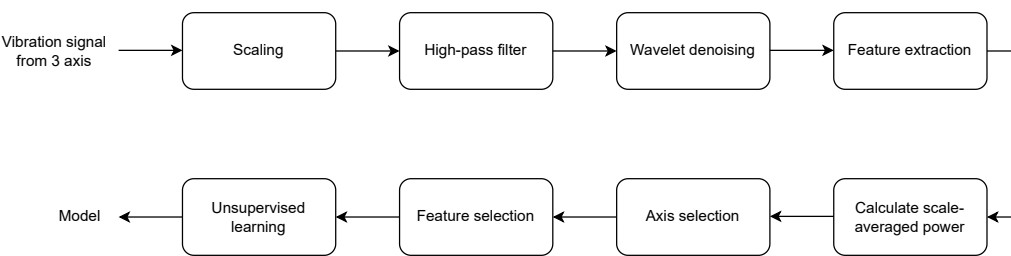

**Figure 5.** Post-processing diagram.

### 2.5. Wavelets

The wavelet transform has been a well-established concept for a considerable amount of time, originating in 1909. It can be broadly categorized into two main types, the continu-

ous wavelet transform (CWT) and the discrete wavelet transform (DWT), each possessing distinct significance in various fields.

CWT is a powerful tool that can be used in time-frequency analysis. It can be thought of as an extension of the short-time Fourier transform (STFT). While the STFT uses fixed-length window functions to analyze a signal, CWT employs wavelet functions that are variable in length and can be adapted to the specific features of the signal being analyzed. The definition of CWT is as follows.

A function $\Psi$ with

$$\int_{\mathbb{R}} \Psi(x)dx = 0 \tag{1}$$

is called a mother wavelet. For every f, $\Psi$ defines the continuous wavelet transform

$$W_\Psi f(a,b) = \int_{\mathbb{R}} f(x)\overline{\Psi(\frac{x-b}{a})}dx \text{ for all a,b } \in \mathbb{R}_+ \times \mathbb{R} \tag{2}$$

where

$$\Psi_{a,b} = \frac{1}{\sqrt{a}}\Psi(\frac{x-b}{a}) \tag{3}$$

The function $\Psi$ is called the wavelet function or the mother wavelet function. It is chosen to be localised at $x = 0$ and at some $\omega = \omega_0 > 0$ (and/or $\omega = -\omega_0$). Here $a$ and $b$ are named the scale factor and the shift factor.

Calculating CWT is computationally expensive. One alternative is to choose a subset of scales and positions to perform the calculations. The selection of scales and positions based on powers of two is a defining feature of discrete wavelet transform (DWT) analysis.

Overall, DWT can be viewed as a form of band-pass filtering that separates a signal into different frequency components, allowing for the extraction of useful information and the creation of a detailed representation of the signal. The definition of DWT is shown below:

$$DWT(a,b) = \frac{1}{\sqrt{a}}\int_{-\infty}^{+\infty} f(x) * \Psi\left(\frac{x-b}{a}\right)dx \tag{4}$$

where

$$a = 2^j, b = k2^j, \left(k,j \in \mathbb{Z}^2\right) \tag{5}$$

Here, $a$ and $b$ are also the scale factor and the shift factor, respectively. Wavelet denoising involves the use of the time-frequency amplitude matrix produced by the wavelet transform. The process of wavelet denoising uses the discrete wavelet transform (DWT) to break down the original signal into its constituent wavelets and obtain the corresponding time-frequency amplitude matrix. To carry out the denoising process, thresholding is applied to the wavelet coefficients associated with the matrix to remove noise and retain the significant components of the signal. Once the thresholding is complete, the signal is reconstructed using the reverse DWT [43]. The application of wavelet denoising is particularly valuable in processing vibration signals, as it can help to remove unwanted noise and enhance the useful features of the signal. Chen et al. [44] demonstrated the effectiveness of this technique by utilizing a wavelet-denoising algorithm to preprocess vibration signals from wind turbines. Chegini et al. [45] introduced a system utilising wavelet denoising in combination with several other signal-processing techniques to improve the accuracy of bearing fault diagnosis. He et al. [46] carried out a study about using a wavelet denoising algorithm with multi-level decompositions to monitor the condition of a heavy-haul railway.

When implementing wavelet denoising, there are several parameters, methods and thresholding techniques that can be chosen. The parameter maximum level of decomposition in wavelet decomposition depends on both the length of the signal (N) and the choice of the wavelet basis. The maximum number of levels for the decomposition of the obtained signals is 21. The levels of the coefficients are known to influence the kurtosis of the signal [47]. The number of levels of the coefficients used in a wavelet decomposition

can have a significant impact on the quality of denoising. A larger number of levels can lead to more aggressive denoising and removal of noise, but may also result in the loss of important signal information and distortion of the signal. In this study, 9 was picked because the features would be extracted from the SAWP. Therefore, some distortion in the time domain may occur and be accepted. The wavelet function used in the wavelet analysis should be carefully selected to reflect the features of the signal in the time domain. With similar reasoning, when the primary interest of the study is the SAWP instead of the time-domain signal then different types of wavelet functions will give similar results [48]. Symlet 4 (sym4), as the most widely used wavelet function, was chosen. To determine the denoising thresholds, the method of empirical Bayesian and median thresholding was selected.

### 2.6. SAWP

SAWP is a measure used in wavelet analysis to quantify the power of a signal at different scales. The SAWP over scales $s_1$ to $s_2$ is defined as follows [48]:

$$\bar{W}_n^2 = \frac{\delta_j \delta_t}{C_\delta} \sum_{j=j_1}^{j_2} \frac{|W_n(s_j)|^2}{s_j} \tag{6}$$

where

$$s_j = s_0 2^{j\delta_j}, j = 0, 1, \ldots, J \tag{7}$$

$$J = \delta j^{-1} log_2(N\delta_t/s_0) \tag{8}$$

$C_\delta$ is a constant that is scale-independent, $\delta_j$ is a factor of scale averaging, $\delta_t$ is the corresponding sampling period and $j_1, \ldots, j_2$ represent the targeting scale range. $s_0$ and $s_j$ represent the smallest and the largest scales, respectively. $W_n(s)$ is known as the CWT of a discrete sequence. Here, N represents the length of the time series [49]. The SAWP will visualise a power burst from the vibration signal when an expected event such as a wheel squat contact or a wheel joint gap contact occurs. This SAWP time series will later be used to extract the SAWP domain features to be input into the machine learning algorithms. The threshold for peak detection was set to $5 \times 10^{-8}$ $g^2$. The threshold was chosen empirically. The SAWP calculated is visualised in Figure 6.

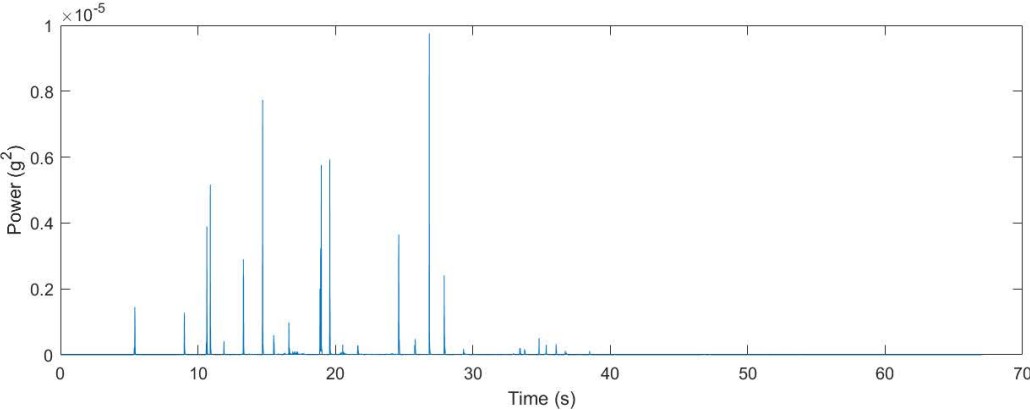

**Figure 6.** The SAWP calculated for a 4 mm test case in the *x* direction.

### 2.7. Unsupervised Machine Learning

In this study, unsupervised learning is used. K-means, DBSCAN and agglomerative hierarchical clustering were utilised to cluster the data. The approach is described and the results are compared.

### 2.7.1. K-Means Clustering

Mac Queen [50] introduced k-means clustering in 1967, which is a simple yet powerful algorithm used to partition a set of observations into $k$ ($\leq n$) sets, where each observation is d-dimensional. The primary goal of the algorithm is to minimize the variance within the clusters. Given an observation set $\{x_1, x_2, x_3..., x_n\}$, the k-means algorithm iteratively assigns each observation to one of the $k$ clusters $S = \{S_1, S_2, S_3, ..., S_k\}$ by minimizing some sort of distance (usually the Euclidean distance) between the observation and the centroid. Mathematically, the objective is to find the following:

$$arg_S min \sum_{i=1}^{k} \sum_{x \in S_i} \|x - \mu_i\|^2 = arg_S min \sum_{i=1}^{k} |S_i| Var S_i \tag{9}$$

where $\mu_i$ is the mean value of all the points in $S_i$. This procedure is equivalent to minimizing the pairwise squared deviations in each cluster:

$$arg_S min \sum_{i=1}^{k} \frac{1}{|S_i|} \sum_{x,y \in S_i} \|x - y\|^2 \tag{10}$$

### 2.7.2. DBSCAN Clustering

The DBSCAN algorithm was first proposed by Martin et al. [51] in 1996. It is a typical density-based clustering algorithm that relies on two parameters, the neighbourhood radius $\varepsilon$ and the minimum number of points minPts, to create a dense area. Two points are considered neighbours if the distance between them is less than or equal to $\varepsilon$. The key ideas are to identify dense regions and to expand them to form clusters. Based on these two parameters, some important concepts are defined below:

A core point is defined as a point that satisfies the criterion of having a minimum of minPts points (including itself) within a radius of $\varepsilon$. On the other hand, a border point is a point that is reachable from a core point but does not meet the minimum density requirement of having at least minPts points within a $\varepsilon$ radius. Any point that does not satisfy either of the two conditions, being a core point or being reachable from a core point, is considered an outlier or noise.

The detailed steps of the algorithm are presented by Chen et al. [52] The main steps are listed as follows:

- Locate all the points within the $\epsilon$ radius of each point.
- Identify the core points that have at least minPts neighbours.
- Find all the core points' connected components in the neighbour graph.
- Assign each non-core point to a cluster if it is within the $\epsilon$ neighbourhood of the cluster.
- Any remaining points are considered outliers or noise.

The advantage of DBSCAN is that it works well with clusters of arbitrary shapes and sizes [25]. Unlike many other unsupervised machine learning algorithms, it does not require setting the parameter of the desired number of clusters.

### 2.7.3. Agglomerative Hierarchical Clustering

An agglomerative hierarchical procedure was coined by Sneath and Sokal in 1973. The approach of this algorithm is constructing a tree structure in bottom-up order. Initially, each object $x$ in the data set is considered as a separate cluster and labelled as a singleton, denoted by $\{x\}$. The clustering process then proceeds iteratively by merging the two closest clusters based on their similarity or distance. For instance, in the case of average linkage,

the distance between two clusters, $C_p$ and $C_q$, can be computed as the mean of the distances between all pairs of points in $C_p$ and $C_q$. [53]

$$D\left(C_p, C_q\right) = \left(\frac{1}{n_p n_q}\right) \sum \left\{ d\left(x_i, x_j\right) \mid x_i \in C_p, x_j \in C_q \right\} \tag{11}$$

Here, $n_p$ and $n_q$ denote the number of elements in clusters $C_p$ and $C_q$, respectively. The distance function $d\left(x_i, x_j\right)$ measures the distance between two data points $x_i$ and $x_j$.

The merged cluster's node level for the joint branches of $C_p$ and $C_q$ is determined using the value of $D\left(C_p, C_q\right)$. The standard approach is monotonic, implying that if a cluster $C$ is incorporated into another cluster $C'$, their respective node levels $LC$ and $LC'$ follow an ascending order.

$$C \subseteq C' \Rightarrow L_c \leq L_{C'} \tag{12}$$

This property ensures the hierarchical tree could be built without branches crossing each other. A simplified description of the steps is as follows:

- Assign each data point its own cluster.
- Compute the similarity information between every pair of clusters (dissimilarity or distance).
- Use a linkage function to group the data into new clusters recursively to build the hierarchical cluster tree, based on the similarity information achieved in the previous step.
- Determine where to cut the hierarchical tree into clusters.

## 3. Results and Discussion

### 3.1. Feature Extraction and Scaling

A few basic features were first extracted from the SAWP. Then, a few more advanced features were created by combining the basic features. A total of ten features were extracted and these features are listed in Table 3. The features are min–max scaled so that the features have a range between 0 to 1. Normalising each feature to the range 0 to 1 range is equivalent to min–max scaling.

**Table 3.** Extracted features from SAWP.

| Feature Number | Feature Level | Description |
| --- | --- | --- |
| 1 | basic | number of peaks |
| 2 | basic | total peak power |
| 3 | basic | mean peak power |
| 4 | basic | root mean square (RMS) |
| 5 | basic | total power |
| 6 | combined | total power/RMS |
| 7 | combined | sum of power > RMS/RMS |
| 8 | combined | number of data points > RMS |
| 9 | combined | total peak power/RMS |
| 10 | combined | mean peak power/RMS |

### 3.2. Axis Selection

According to the calculations, the signals in the $x$ and $z$ directions are similar and it is reasonable to use the signals from those two directions as individual trials. This expanded the data set from nine test trials to 18. The signal from the y direction was omitted from further processing.

### 3.3. Feature Selection

The Laplacian score of each feature was calculated and sorted to decide the importance of different features. The absolute minimum number of features that can be selected is one. However, this is not practical and any noise in that feature might cause the algorithm to not

function. At the same time, including more features also requires more data points. In this study, the data available are limited so a minimum set of features is preferable. Therefore, two features were decided to be selected. The top two features were feature 1, the number of peaks, and feature 5, the total power. These two features were be utilised as input to the unsupervised clustering algorithms later.

### 3.4. K-Means Clustering

The normalised features were fed to the k-means clustering algorithm. In general, the k value should be decided by applying the elbow method. The elbow method was tested. However, the method suggested the optimal cluster number should be nine. The reason behind it is that the data set is small. On the other hand, using silhouette scores as measurements implies that four clusters is optimal. Since it is known that there were three different categories, the k value was set to 3. The results of the clustering are shown in Figure 7. By checking the label, all the data points with the same label were clustered together. From bottom left to top right, the cluster regions represent the no-squat, 1 mm deep squats and 4 mm deep squats cases. It is also possible to see from the figure that the 4 mm data are well-separated from the other two cases while the 1 mm data are close to the no-squat case. The corresponding confusion matrix achieved using the actual class labels is shown in Table 4. The achieved accuracy is 100%.

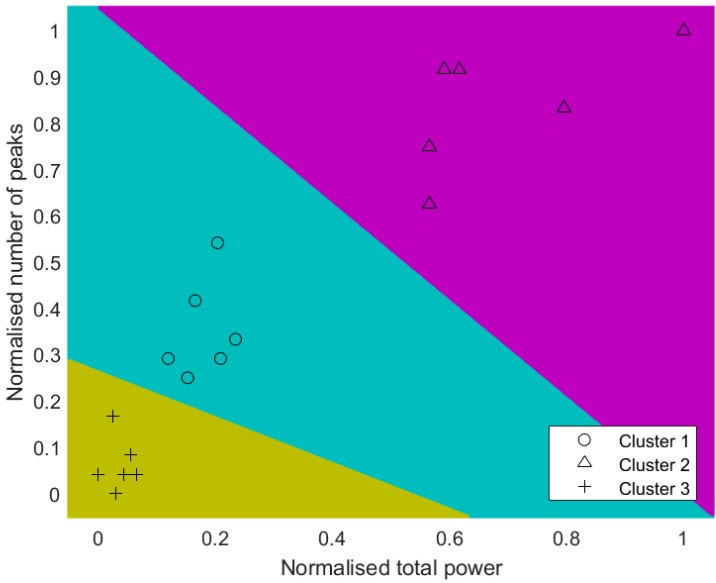

**Figure 7.** K-means decision boundaries and clusters.

**Table 4.** Confusion matrix for k-means algorithm.

|  |  | **Predicted Cluster** |  |  |
| --- | --- | --- | --- | --- |
|  |  | **0 mm** | **1 mm** | **4 mm** |
|  | 0 mm | 6 | 0 | 0 |
| Actual label | 1 mm | 0 | 6 | 0 |
|  | 4 mm | 0 | 0 | 6 |

### 3.5. DBSCAN Clustering

The normalised features were fed to a DBSCAN clustering algorithm. There are two parameters that need to be decided, namely, the epsilon value and the minimum number of samples. Usually, the minimum number of samples is set to one plus the number of dimensions of the input data [51], which is three in our case. The epsilon value is estimated using the k-distance graph as shown in Figure 8. The knee appears to be around 0.14;

therefore, the value of epsilon is set to 0.14. The result of the clustering is visualised in Figure 9. By comparing the labels, it can be seen that the no-squat case and the 1 mm squat depth case are grouped correctly. All data points of the 4 mm squat depth case were clustered as anomalies. DBSCAN is density-based, so a small data set will be challenging since it is more difficult to find neighbours, especially when the cluster has a larger variance. The corresponding confusion matrix achieved using the actual class labels to compare with the predicted clusters is shown in Table 5. The achieved accuracy is 66.7%.

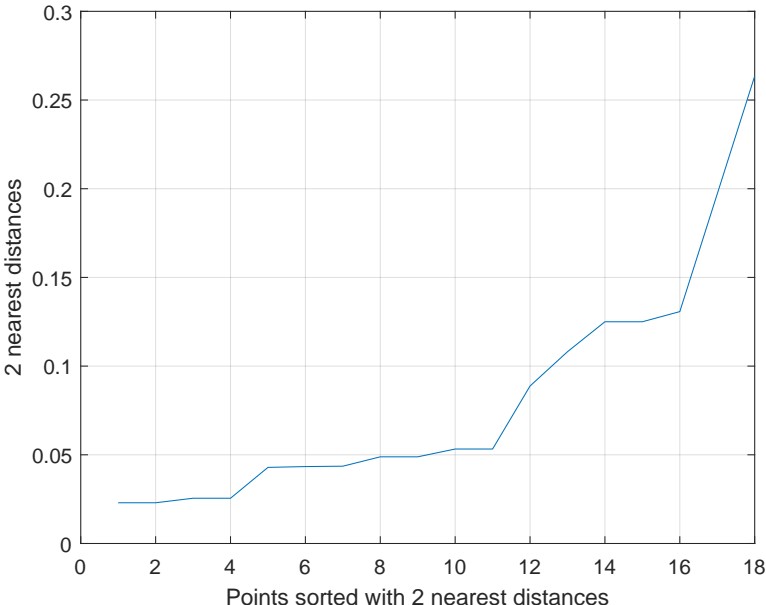

**Figure 8.** K-distance graph with k set to 2.

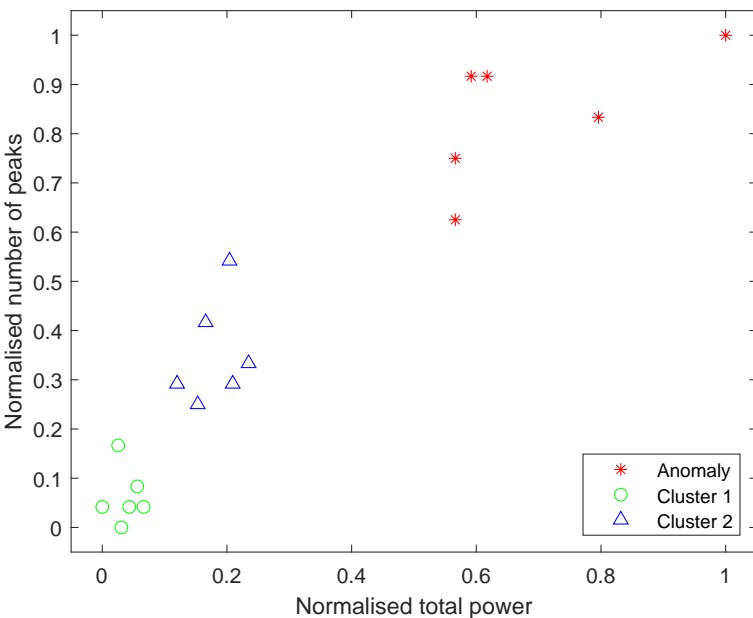

**Figure 9.** DBSCAN clusters with noise/anomaly.

**Table 5.** Confusion matrix for DBSCAN algorithm.

|  |  | Predicted Cluster | | | |
|---|---|---|---|---|---|
|  |  | **0 mm** | **1 mm** | **4 mm** | **Noise** |
| Actual label | 0 mm | 6 | 0 | 0 | 0 |
|  | 1 mm | 0 | 6 | 0 | 0 |
|  | 4 mm | 0 | 0 | 0 | 6 |
|  | noise | 0 | 0 | 0 | 0 |

### 3.6. Agglomerative Hierarchical Clustering

The normalised features were fed to an agglomerative hierarchical clustering algorithm. A few parameters can be set for the algorithm. Since the data set is small, the depth of the hierarchy was set to four and the maximum number of clusters was set to three. The dendrogram generated with the algorithm using the data is shown in Figure 10. The results of the clustering are visualised in Figure 11. By checking the labels, all the data points with the same label were clustered together. From left to right, the clusters represent the no-squat, 1 mm deep squats and 4 mm deep squats cases. The corresponding confusion matrix achieved by comparing the predicted clusters with the actual class labels is shown in Table 6. The achieved accuracy is 100%.

**Table 6.** Confusion matrix for agglomerative hierarchical algorithm.

|  |  | Predicted Cluster | | |
|---|---|---|---|---|
|  |  | **0 mm** | **1 mm** | **4 mm** |
| Actual label | 0 mm | 6 | 0 | 0 |
|  | 1 mm | 0 | 6 | 0 |
|  | 4 mm | 0 | 0 | 6 |

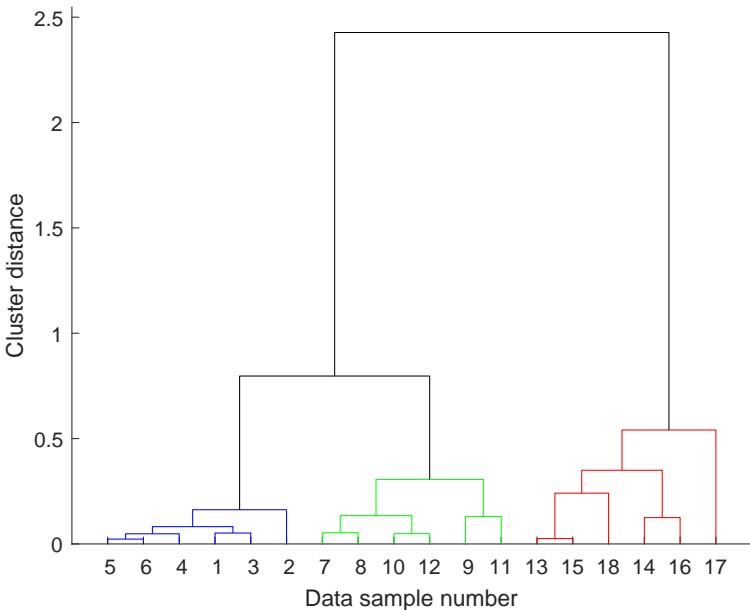

**Figure 10.** Dendrogram showing the three clusters.

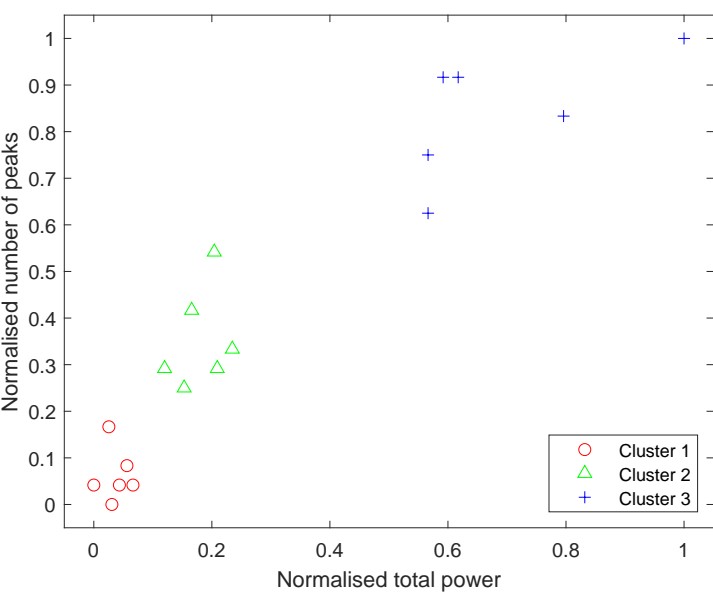

**Figure 11.** Agglomerative hierarchical clusters.

## 4. Conclusions and Future Works

This study shows it is possible to combine the proposed method to extract and select two important features and unsupervised machine learning algorithms to cluster squats of different levels. In detail, the following conclusions are drawn:

- The presented signal processing method is effective and promising to extract useful information from the vibration signal.
- It is possible to only utilise features from SAWP from the vibration signal to identify different degrees of squat defects of the S&Cs.
- The number of peaks and the total power are the two most important features that can be utilised to estimate the squat levels.
- Both k-means and agglomerative hierarchical clustering provide similar good results.
- The DBSCAN clustering encounters some challenges and clusters the 4 mm depth case as anomalies; therefore, it is not suitable for using this algorithm to process such a data set.

The conclusions above show it is promising to utilise the features extracted from SWAP time series to determine the level of squats with unsupervised learning algorithms such as k-means or agglomerative hierarchical clustering. The presented method of measuring the vibration signals at the point machine could also possibly be used for detecting other types of defects in the S&C such as wear, point machinery defects and vehicle wheel defects such as wheel flats. There are some limitations of this study. First, the experiments were performed under a low-speed condition (<2 m/s) with a constant load. Second, the data set size is small. Future studies will address these two limitations.

It is possible to perform an extra parametric study similar to what Molodova et al. [36] did and study the influence of different speeds and loads on the collected vibration signals. Combining the approach presented in this study and the emerging technique of federated learning could potentially construct a nationwide condition-monitoring system for S&Cs. Another possible future study is to collect more data from a real railway system to further verify the method presented in this study. One last future study could be to study the influence of parameters such as different train types, variant load and different operating speeds on the vibration signals and further improve the current algorithm to include these factors.

**Author Contributions:** Conceptualization, Y.Z. and J.L.; methodology, Y.Z., M.R. and P.C.; software, Y.Z. and formal analysis, Y.Z.; data acquisition, J.L. writing—original draft preparation, Y.Z.; writing—review and editing, Y.Z., M.R., J.L. and P.C.; visualization, Y.Z. All authors have read and agreed to the published version of the manuscript.

**Funding:** This research received no external funding.

**Institutional Review Board Statement:** Not applicable.

**Informed Consent Statement:** Not applicable.

**Data Availability Statement:** Not applicable.

**Acknowledgments:** Great thanks to Taoufik Najeh for help designing and building of the test rig and collecting all data.

**Conflicts of Interest:** The authors declare that there are no conflict of interest.

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
