# Peer review of "Squat Detection and Estimation for Railway Switches and Crossings Utilising Unsupervised Machine Learning"

_applsci, doi:10.3390/app13095376_

Round 1

Reviewer 1 Report

The installation of vibration sensors on railway switches to monitor and analyze squat defect is appropriate for switch systems. The research work has some practical value through on-site testing and analysis. But the paper needs further revision in the following aspects.

1. Whether the load and velocity changes acting on the switch affect the test results. The speed of this test should be low and the load should be small.

2. Comparison and analysis of the difference between the test results of the sensors placed on the blade and on the rod of point machine.

3. Explain the vibration sensors in the x, y, and z directions required for this test.

4. Whether changes in the position of squat defects affect test results?

5. Add some content of the test data analysis process.

Author Response

Our reply is attached.

Reviewer 2 Report

The authors proposed a method for monitoring switches and crossings. They have found two features and applying unsupervised monitoring systems different squats defects can be detected on switches and crossings.

Please, enlarge the abstract indicating the two features found and describe the main challenges.

The state of the art is enough.

In results section, a clear quantification of system success in detection and classification of squats defects is needed.

There is not discussion section where authors include comparison with other results on the studied field.

Please, add in conclusions section the limitations of the study and others applications of the methodology (extracting features,e.g.)

Review English grammar and style for the full document, also there are typos (Line 16, county instead of country. L174, Dtata….)

Author Response

The reply is attached.

Reviewer 3 Report

The paper presents a railway squat detection and estimation method.  The following points should be addressed before publication. 1. The introduction seems too board. Some highly relevant works are ignored, such as the Automated and adaptive ridge extraction, and the Time-frequency ridge estimation: An effective tool for gear and bearing fault diagnosis at time-varying speeds. 2. The samples used for demonstration is too small, especially for machine learning. 3. The proposed method should be compared with the related methods to verify its superiority.

Round 2

Reviewer 3 Report

I have no more comments. The paper can be published.